# High Risk Clone: A Proposal of Criteria Adapted to the One Health Context with Application to Enterotoxigenic *Escherichia coli* in the Pig Population

**DOI:** 10.3390/antibiotics10030244

**Published:** 2021-02-28

**Authors:** Maud de Lagarde, Ghyslaine Vanier, Julie Arsenault, John Morris Fairbrother

**Affiliations:** 1OIE Reference Laboratory for *Escherichia coli*, Faculty of Veterinary Medicine, Université de Montréal, Saint-Hyacinthe, QC J2S2M2, Canada; maud.de.lagarde@umontreal.ca (M.d.L.); ghyslaine.vanier@umontreal.ca (G.V.); 2Swine and Poultry Infectious Research Center (CRIPA-FQRNT), Faculty of Veterinary Medicine, Université de Montréal, Saint-Hyacinthe, QC J2S2M2, Canada; julie.arsenault@umontreal.ca; 3Groupe de Recherche en Epidémiologie des Zoonoses et Santé Publique (GREZOSP), Faculty of Veterinary Medicine, Université de Montréal, Saint-Hyacinthe, QC J2S2M2, Canada

**Keywords:** *Escherichia coli*, ETEC:F4, fluoroquinolones non-susceptibility, antimicrobial resistance, pigs, genomics, multidrug resistance, North America

## Abstract

The definition of a high risk clone for antibiotic resistance dissemination was initially established for human medicine. We propose a revised definition of a high risk clone adapted to the One Health context. Then, we applied our criteria to a cluster of enrofloxacin non susceptible ETEC:F4 isolates which emerged in 2013 in diseased pigs in Quebec. The whole genomes of 183 ETEC:F4 strains isolated in Quebec from 1990 to 2018 were sequenced. The presence of virulence and resistance genes and replicons was examined in 173 isolates. Maximum likelihood phylogenetic trees were constructed based on SNP data and clones were identified using a set of predefined criteria. The strains belonging to the clonal lineage ST100/O149:H10 isolated in Quebec in 2013 or later were compared to ETEC:F4 whole genome sequences available in GenBank. Prior to 2000, ETEC:F4 isolates from pigs in Quebec were mostly ST90 and belonged to several serotypes. After 2000, the isolates were mostly ST100/O149:H10. In this article, we demonstrated the presence of a ETEC:F4 high risk clone. This clone (1) emerged in 2013, (2) is multidrug resistant, (3) has a widespread distribution over North America and was able to persist several months on farms, and (4) possesses specific virulence genes. It is crucial to detect and characterize high risk clones in animal populations to increase our understanding of their emergence and their dissemination.

## 1. Introduction

It is now well recognized that antimicrobial resistance threatens environmental, animal, and public health [1]. The dissemination of antimicrobial resistance genes through human populations, animal populations, and the environment is no longer a subject for debate [2]. There are several known mechanisms of such dissemination that can be broadly classified as horizontal gene transfer (HGT), mainly through mobile genetic elements, or clonal dissemination. The term “high risk clone” emerged around 2011 [3] and has been used to describe bacterial clones that enhance the dissemination of antibiotic resistance [4,5]. These clones represent a major concern not only because they pose a substantial challenge for the treatment of patients but also because they are very efficient vehicles for mobile genetic elements carrying antimicrobial genes, and therefore promote the spread of these genes. The dissemination of ST131-H30-Rx *Escherichia coli* causing mainly urinary tract infection and septicemia in humans, which is resistant to fluoroquinolones and third generation cephalosporins, is one of the best examples of this problem [6]. In its current definition, the term “high risk clone” is mainly adapted for human pathogens, whereas antimicrobial resistance should be addressed using a more global approach [7]. Therefore, it may be timely to revisit the present criteria [8] and to adapt them to the One Health concept [9].

The factors driving the emergence and the dissemination capacity of high risk clones are not well understood. Several fitness characteristics, particularly the speed of replication, have been associated with high resistance to fluoroquinolones [10] and have been demonstrated to confer growth advantage to several clones, particularly in extended-spectrum β-lactamase (ESBL) producing *E. coli* [11]. Therefore, fluoroquinolone use leading to fluoroquinolone resistance could contribute to the selection and the spread of high risk clones [12,13].

Enterotoxigenic *E. coli* (ETEC) causes diarrhea in piglets during the neonatal and post-weaning periods, resulting in important economic losses for the swine industry [14]. ETEC strains produce one or several enterotoxins, which induce the secretion by epithelial cells of water and electrolytes into the intestinal lumen. These ETEC strains colonize the host via fimbriae which adhere to the intestinal mucosa. ETEC associated with diarrhea in young pigs most commonly possess F4 (K88) or F18 fimbriae, although geographic variations have been reported [15]. ETEC isolates are grouped into pathovirotypes based on the presence of the different fimbriae and one or more of the enterotoxins heat labile toxin (LT) and heat stable toxins (STa and STb). Pigs with diarrhea associated with ETEC have been commonly treated with ampicillin (alone or in combination with clavulanic acid), trimethoprim, sulfonamides, aminoglycosides (i.e., neomycin), ceftiofur, spectinomycin, or enrofloxacin, depending on the country [15]. In a previous article, we reported the presence of a cluster of ETEC:F4 isolates sharing at least 55% of similarity based on their pulse field gel electrophoresis (PFGE) profile in diseased pigs in Quebec, which emerged in 2013 [16]. Most of these isolates were LT:STb:STa:F4 and non-susceptible (intermediate or resistant) to enrofloxacin. This antimicrobial has been used off-label in Quebec since the 2000s. In addition, fluoroquinolone non-susceptibility in *E. coli* isolated from diseased pigs has been observed in other countries [17,18] and has been associated with enrofloxacin treatment previously in herds [19]. Moreover, clonality has been demonstrated among ETEC isolates causing disease in pigs in the USA [20]. Our hypothesis is that a high risk fluoroquinolone non-susceptible pathogenic ETEC clone has emerged in pigs in various countries, thus representing a threat for swine and public health. The objectives of this study were to (1) propose a new definition including revised criteria for a high risk clone adapted to the One Health context, (2) further characterize ETEC:F4 isolates sampled in Quebec, using a genomic approach, and (3) compare these isolates with other ETEC:F4 worldwide both phylogenetically and with respect to the presence of virulence and antimicrobial resistance genes and mobile genetic elements, in order to assess the presence of a high risk clone based on well-defined criteria.

The detection and surveillance of these putative new clonal isolates is crucial to identify and develop control strategies in the field.

## 2. Results

### 2.1. Proposal of Criteria for a High Risk Clone in the One Health Context

Until now, clones have qualified as being “high risk” when they have met 6 criteria: (i) worldwide spread, (ii) carriage of multiple antimicrobial characteristics, (iii) efficient colonisation and persistence in host, (iv) effective transmission, (v) increased pathogenicity, and (vi) increased virulence causing recurrent infection [6,21]. Other important aspects, such as the emergence of the pathogen, environmental persistence, and zoonotic transmission, were not included in these characteristics. Therefore, we propose a list of new criteria, presented in Table 1, to adapt the “high risk” clone definition to a One-Health context. According to our revised criteria, a clone qualifies to be high risk for antibiotic resistance dissemination if (1) it is emergent, the definition of emergence of a pathogen is as described in Wang et al. [22] and is based on OIE recommendations; (2) it carries multiple resistance genes associated with phenotypic resistance; (3) it has a high capacity of dissemination, the definition of persistence was adapted from Falkow et al. [23] and the concept of environmentally maintained source is described in Blackburn et al. [24]; and (4) it is highly pathogenic. The definitions of these criteria are detailed in Table 1.

### 2.2. Identification of Clonal Lineages and Clones

We sequenced a total of 183 isolates originating from diseased pigs in Quebec, Canada, between 1990 and 2018, randomly selected in each cluster of a preliminary PFGE study or from the APZEC database. After removing 10 isolates due to unsatisfactory raw data or assembling quality, 36 isolates from 1990 to 2012 as well as 21, 21, 22, 26, 24, and 23 isolates for each year from 2013 to 2018, respectively, were retained. The sequencing quality information about the 173 isolates is available in Appendix A.

A total of 3,366,150 positions were found in all analyzed genomes. The percentage of reference genome covered by all isolates was 61.95% with a reference genome size of 5,433,365 pb (ECL19664). After the removal of the recombination signal with the Gubbins software, the total number of SNPs used to build the tree was 19,229. The resulting SNP tree of the 173 isolates is shown in Figure 1A. Based on the definitions described in the Materials and Method, 6 clonal lineages (A, B, C, D, E, and F) and 6 clones (A1, A2, A3, A4, B1 and C1) were identified. One sub-clone within the clone A1, designated A1-sub was also detected. Two singletons were also identified (ECL01947 and ECL20395). The SNP_max_ of each group is presented in detail in Appendix A.

In the 1990s and the early 2000s, clonal lineages D, E, and F were predominant (Figure 1A). Clonal lineage D was the oldest, 10/12 isolates being obtained between 1990 and 1995. All isolates (D, E and F) were from the phylogroup C, MSLT ST90 and presented the gene *fimH54*. However, isolates of the three clonal lineages belonged to different serotypes, respectively O149:H43, O149:H19 and O159:H19 and no clone was detected within these lineages according to the set of predefined criteria.

The predominant clonal lineage A, comprising 129/173 isolates, was first observed in 1995. The phylogeny of these 129 isolates has been enlarged and is presented in more detail in Figure 1B. These 129 isolates are O149:H10, ST100, phylogroup A and possess no known *fimH* gene. Within this clonal lineage, 4 clones (A1, A2, A3 and A4) were identified. Isolates from the clones A2 and A3 were identified around the same period (1995 to 2014), prior to isolates from the clones A1 and A4. Isolates from the clone A1 were first observed in 2013 and have continued to be observed over the 5-year period until the end of the study. The subclone (A1-sub) was composed of isolates sampled between 2015 and 2018. Isolates belonging to the clone A4 were isolated during the same period as the clone A1 (between 2013 and 2018) but represent a much smaller proportion of isolates from the clonal lineage A.

The clonal lineage B/clone B1, including 11 isolates, was first observed in 2009, and continued to be found until the end of the study period (Figure 1A). These isolates are O138:H10, ST100, phylogroup A and present no known *fimH* gene.

More recently, in the last 2 years of the study period, the clonal lineage C/clone C1, belonging to a new sequence type, has been observed. These isolates are O23:H37, ST772, phylogroup A and possess the gene *fimH54*.

In summary, ETEC:F4 isolates in Quebec in the early 1990s were predominantly ST90, belonging to 3 different serotypes and carrying the *fimH54* gene. In the mid-1990s, isolates belonging to ST100 appeared and became predominant. They belong to 2 main O serotypes and carry no *fimH* gene. The clonal lineage A (ST100/O149:H10) is the most widespread and is composed of several clones. In the late 2010s, isolates belonging to another clonal lineage have appeared. They belong to the ST772 and the serotype O23:H37 and constitute a clone.

### 2.3. Virulence Gene Profiles of the Different Clonal Lineages

The virulence gene profiles of the isolates were established based on the detection of these genes in the VirulenceFinder and VFDB databases. ETEC:F4 isolates of the same clonal lineage mostly belonged to the same virotype (Figure 1A,B). All isolates from clonal lineages D, E, F and B1 (apart from one in B1) were LT:STb:F4 and isolates from C1 were STa:STb:F4. Isolates from the clonal lineage A belonged to the virotype LT:STb:F4 except for isolates from clones A1 and A2 which possessed various combinations of enterotoxins, most isolates being LT:STb:STa:F4 (49/80 and 9/11 respectively).

Clonal lineage A differed from the other lineages by the absence of the group of genes coding for the siderophore Yersiniabactin (*irp, fyuA* and *ybt* family genes), by the presence of genes involved in the type VI secretion system (*aec* family genes) and the presence of genes coding for elements of the *E. coli* common pilus (ECP) and of the hemorrhagic *E. coli* pilus (HEP). Clonal lineage A was also distinguished by the presence of *ompA* (interacts with specific receptors for initiating the pathogenic process), *rpoS* (involved in the regulation of the stress response) and *tar/cheM* (involved in the chemotactic response) (Table 2 and Appendix A). The *ehxA* gene encoding enterohemolysin (associated with haemolysin secretion) was detected in all isolates belonging to the clonal lineage C but in none of the isolates belonging to the other clonal lineages. The *iha* gene (encoding for a homologue adhesin (IrgA)), was detected in all isolates belonging to the clonal lineage B but in none of the isolates belonging to the other clonal lineages. The *lfpA* gene (encoding for a fimbrial major protein) was present in all isolates belonging to clonal lineages D, E and F but none of the isolates belonging to other clonal lineages. The porcine attaching-effacing associated gene (*paa*) was detected only in isolates belonging to the clonal lineage A. However, its presence was not systematic (83/129) and was not specific to a clone. There was no virulence gene identified exclusively in isolates belonging to a particular clone and in all isolates from this clone. All these data are available in Appendix A.

### 2.4. Resistance Gene, Multidrug Resistance (MDR) and Replicon Profiles of the Different Clonal Lineages

The resistance gene and replicon profiles of the isolates were established based on the detection of these genes in the ResFinder, PlasmidFinder and ARG-Annot databases. Isolates from clonal lineage E (*n* = 10) were mostly non-susceptible to tetracycline (9/10) and ampicillin (7/10) and showed no or little non-susceptibility to enrofloxacin (0/10), cephalosporins (1/10), florfenicol (3/10), TMS (4/10), and aminoglycosides (5/10) (Appendix A). Genes detected and likely responsible for these non-susceptibilities were *tet(B)* for tetracycline, *bla_TEM-1B_* for ampicillin, *cmlA1* or *catA1* for florfenicol, *sul1* and *drf1* for TMS, and *aad* or *aph* for aminoglycosides.

Half of the isolates from clonal lineage D (*n* = 12) were non-susceptible to tetracycline (6/12), otherwise they were infrequently non-susceptible to ampicillin (2/12), enrofloxacin (0/12), cephalosporins (1/12), florfenicol (2/12) and TMS (2/12), and aminoglycosides (6/12) (Appendix A). Genes detected and likely responsible for these non-susceptibilities were *tet(A)* or *tet(B)* for tetracycline, *bla_TEM-1B_* for ampicillin, *cmlA1* or *catA1* for florfenicol, *sul1* and *drf1* for TMS, and *aad* for aminoglycosides (Appendix A). IncFIB and IncFII replicons were present in isolates from both clonal lineages D and E. However, the IncFIC replicon was only present in the clonal lineage E (Appendix A). One isolate in D carried a *bla_IMP_* (responsible for carbapenemase activity) (Appendix A).

Isolates from clonal lineage B1 (*n* = 11) were mostly non-susceptible to tetracycline (11/11), to TMS (11/11), to aminoglycosides (9/11), and to ampicillin (6/11) and infrequently non-susceptible to enrofloxacin (0/11), cephalosporins (5/11), and florfenicol (2/11) (Appendix A). Genes detected and likely responsible for these non-susceptibilities were *tet(A)* and *tet(M)* for tetracycline, *bla_TEM-1B_* for ampicillin, *cmlA1* or *catA1* for florfenicol, *sul1* or *sul2* and *drf1* for TMS, and *aad* or *aph* for aminoglycosides (Appendix A). The plasmid profile was also similar among B1 isolates with the presence of IncI1 and IncFIB (Appendix A).

Isolates from clonal lineage A (*n* = 129) were mostly non-susceptible to tetracycline (115/129) and to ampicillin (113/129). They were infrequently non-susceptible to TMS (55/129), to aminoglycosides (61/129), to cephalosporins (31/129), and to florfenicol (40/129) (Appendix A). Concerning the enrofloxacin non-susceptibility there was a clear separation between isolates belonging to the clone A1, which were mostly non-susceptible (74/80) and those of the clonal lineage A not belonging to the clone A1 which were very rarely non-susceptible (1/49). Therefore, isolates belonging to A1 were mostly MDR (71/80).

Isolates from the clonal lineage A carried two mutations, *parC* (E62K) and *gyrA* (N652H), classified as «unknown» (meaning that the phenotypic significance is not established) in ResFinder. All isolates from the clone A1 carried these two mutations as well as two other mutations in the *parC* (S80I) and *gyrA* (S83L) genes, classified as «known» in ResFinder (meaning that these mutations were associated with a non susceptible phenotype). Isolates from the clonal lineage A also carried the *tet(A)* (109/129) or the *tet(B)* (11/129) which confer tetracycline resistance (Figure 1B). The *tet(B)* gene was mainly present in isolates from the clone A4. Except for isolates from the clone A3, they also carried the *bla_TEM-1_* (104/129) gene which confers ampicillin resistance. All A1 isolates carried the replicon FIB(K) which was found only in two other isolates within the clonal lineage A (one in the clone A2 and one in an isolate belonging to no clone (Appendix A)).

Isolates from the most recently occurring clonal lineage C1 (*n* = 7) were all non-susceptible to tetracycline (7/7), to aminoglycosides (7/7), and to TMS (7/7), therefore, they were all MDR. Moreover, they were mostly non-susceptible to florfenicol (6/7) and infrequently non-susceptible to enrofloxacin (1/7), cephalosporins (3/7), and ampicillin (3/7) (Appendix A). Genes detected and likely responsible for these non-susceptibilities were *aah* gene for aminoglycosides, *tet(A)* for tetracycline resistance and *sul3* and *dfrA12* for TMS. It is noticeable that one of the isolates was non-susceptible to quinolones, although this isolate carried a *qnr* gene. Also, despite the presence of *cmlA1* (supposedly conferring resistance to phenicols) two of these isolates were susceptible to florfenicol. Two isolates carried the *bla_CTX-M-1_* gene and were resistant to ceftiofur. The plasmid profile was also similar among C1 isolates, IncFIB(K) and (PB71) being always present.

Overall, the AmpC gene *bla_CMY-2_* was detected in 23/173 isolates, belonging to clonal lineages A (22/23) and E (1/23). In our isolates, this gene was first detected in 2005. It is interesting to note that when the *bla_CMY-2_* was present, the replicon IncA/C was also present, suggesting a link between them (data not shown). In our isolates, the ESBL gene *bla_CTX-M-1_* was first detected in 2013 and was found in 15/173 isolates belonging to recent clonal lineages (A(9/15), B(4/15) and C(2/15)). However, neither *bla_CMY-2_* nor *bla_CTX-M-1_* was specific to a clone within the clonal lineage A.

### 2.5. Mortality Risk, Production Phase of Affected Pigs, and Persistence on Farm of the Different Clonal Lineages

Necropsy reports were available for 106 cases. Overall, mortality was observed in 47% of these cases. The risk of mortality was significantly (*p* = 0.003) higher in cases infected with isolates from clonal lineage A (54%) compared to clonal lineage B (0%) but was not associated with other clonal lineages. Within clonal lineage A, no significant difference in the mortality risk was observed between cases infected with isolates from different clones (*p* = 0.09) (see Appendix A).

Information on the phase of production of the pigs was available for 158 cases. Overall, most cases of ETEC:F4 infection (73%) were detected in pigs in the weaning phase in nurseries. No statistically significant association was found between clonal lineage and phase of production (*p* = 0.5), nor between clones and phase of production within the clonal lineage A (*p* = 0.7) (Appendix A).

In 2 farms located in Montérégie, Québec, 2 isolates belonging to the clone A1 were identified 6 and 8 months apart, respectively. In both cases, the farms were housing two stages of production: a farrowing barn and a nursery. For both cases, the first isolate was identified in the farrowing barn and the second isolate was identified in the nursery, strongly suggesting that the clone persisted on farm. The type of cleaning and rotation were not known for either farm. No other pair of isolates was sampled at different times on the same site, according to the available information.

### 2.6. Presence of Isolates Belonging to the Clonal Lineage A in North America

Eighty-seven whole genome sequences in Enterobase were described as ETEC and ST100 isolated from diseased pigs between 2013 and 2018 and were available in GenBank. Among these, 7 were duplicates and were removed from our analysis. The 112 isolates belonging to the clonal lineage A and sampled in Quebec after 2013 and the 80 isolates identified in Enterobase were used for the second phylogenetic analysis. All the isolates described in Enterobase were confirmed to belong to the ST100, O149:H10 serotype, phylogroup A, and to carry no *fimH* gene in silico. These isolates originated from various states of the USA except for one from a sample in Switzerland. The available metadata for these isolates are presented in Appendix A.

A total of 3,675,678 positions were found in all analyzed genomes. The percentage of reference genome covered by all isolates was 67.65% with a size of reference genome of 5,433,365 pb (the same isolate was used as reference). After removal of the recombination signal the total number of SNPs used to build the tree was 3119. As expected, because isolates belonged to the same clonal lineage, the number of differing SNPs was low (min = 0, max = 1007, median =38), indicating that isolates were very similar. The resulting SNP tree of the 192 isolates is illustrated in Figure 2 and distance matrix is presented in Appendix A.

Two clones were identified and designated as A-I and A-II and one subclone designated A-I-sub. The clone A-I (Figure 2 and Appendix A) comprised 150 isolates, originating from both Canada and USA. All isolates from the clone A1 and only isolates from the clone A1 in the previous analysis belonged to the clone A-I. Therefore, we will use the designation clone A-I to refer to the isolates from both analyses. The clone A-II in this analysis comprised the same isolates as those of the clone A4 in the previous phylogenetic analysis.

Clone A-I isolates possessed various combinations of enterotoxin virulence genes but were mainly LT:STb:STa:F4 (98/150). These virulence profiles and those identified in Table 2 did not differ between isolates of clone A-I and those of the clonal lineage A.

All isolates from clone A-I carried a mutation in each of the *parC* and *gyrA* genes (S80I and S83L respectively), classified as «known» in ResFinder, and a mutation in each of the *parC* and *gyrA* genes (E62K and N652H, respectively), classified as «unknown» in ResFinder. In 16 isolates, an additional mutation was detected in the *gyrA* gene (Figure 2 and Appendix A). Fourteen of these 16 isolates were sampled in the USA and 2/16 in Quebec. Most of the clone A-I isolates also carried *bla_TEM-1_* (142/150) and *tet(A)* (134/150); however, susceptibility phenotype results were not present in the metadata available in Enterobase.

In the isolates sampled in the USA, the ampC gene *bla_CMY-2_* was detected in 16/80 isolates and the ESBL gene *bla_CTX-M-1_* was detected in only 1/80 isolates. The A-I isolates also carried replicons of the incompatibility group FIB (Appendix A).

### 2.7. Application of the Criteria for “High Risk” Clone to the Clones A-I, A2, A3, A4, C1 and B1

As illustrated in Table 3, the clone A-I emerged in 2013, which is multidrug resistant and carries several genes associated with multidrug resistance, has disseminated efficiently throughout North America and is associated with severe diarrhea and/or sudden death. Therefore, the clone A-I qualifies as a high risk clone. Other clones in the clonal lineage A (A2, A3, A4) are not consistently MDR and did not disseminate as efficiently as the clone A-I. The clone B1 is not consistently MDR and does not predominate currently. The clone C1 emerged around the year 2014, is MDR and pathogenic. Until now, there is no proof that it has disseminated throughout North America.

## 3. Discussion

The main objective of this study was to redefine the term “high risk clone”. Indeed, bacteria (and other microorganisms) lodge in multiple ecological niches, such as humans, animals, or the environment. A comprehensive and integrated approach considering the complex interrelationships between these niches is needed in the development of global strategies tackling antimicrobial resistance [25]. In this regard, the One Health approach, defined as “the collaborative effort of multiple health science professions, together with their related disciplines and institutions -working locally, nationally, and globally -to attain optimal health for people, domestic animals, wildlife, plants, and environment” [26] is essential to investigate all the multifaceted mechanisms for antimicrobial resistance dissemination. Yet, the current definition of a high risk clone is mainly adapted to human medicine alone [6,21]. The second objective was to determine if enrofloxacin-non-susceptible ETEC:F4 isolates from diseased pigs isolated in Quebec belonged to a high risk clone.

Our analytical approach permitted us to examine the phylogenetic variation of ETEC:F4 isolates from diseased pigs with time in Quebec. Prior to 2000, ETEC:F4 isolates were mostly ST90 and belonged to several serotypes. On the other hand, after 2000, isolates were mostly ST100, belonging to only one serotype, O149:H10, and carrying no *fimH* gene. We designated these isolates as the clonal lineage A. Within this clonal lineage we identified several clones and subclones. In addition, in a second phylogenetic analysis, we compared isolates from the clonal lineage A to other ST100/O149:H19 isolates sampled in other geographical locations. All enrofloxacin-non-susceptible isolates belonged to a clone designed as A-I. As the clone A-I of the second analysis included all isolates from the clone A1 of the first analysis, we will use the designation clone A-I to refer to the isolates from both analyses. In the next paragraphs we will discuss each revised criterion for a high risk clone and its application to the clone A-I.

We demonstrated that the clone A-I appeared in 2013 and became predominant in 2015 (data from this study and [16]), consequently meeting the emergence criteria (cf Table 1). The clonality of ETEC isolates from various states of the USA and belonging to clonal lineage A had already been demonstrated [20]. In addition, we demonstrated the international distribution of the clone A-I, as it is also present in Canada, at least in Quebec. Thus, we demonstrated that isolates of the clone A-I spread in Canada and in the USA, possibly via the frequent displacement of pigs between the two countries. As an illustration, in 2015, Canada exported almost 6 million live hogs to the USA for feeding or harvest and imported 528 million pounds of USA pork meat (https://www.pork.org/facts/stats/canadian-statistics/ accessed on 28 February 2021). A wider distribution of this clone was difficult to assess, as whole genome sequences of *E. coli* from other continents were not available in Enterobase within the study period. Indeed, the common use of whole genome sequencing occurred later in veterinary medicine than in human medicine, due mainly to its high cost. Consequently, veterinary isolates are less frequently available in databases. In particular, it would have been very interesting to include sequences from isolates in Japan where PFGE-based cluster lineages demonstrating fluoroquinolone non-susceptibility have been reported [18]. This wide distribution is the result of a wide capacity of dissemination (cf Table 1). Yet, this capacity of dissemination can be driven by several means. In our definition of a high risk clone, we wished to include all factors of importance that could increase the likelihood of clonal dissemination to human and/or animal populations, to match as many field situations as possible. Therefore, we established three inherent sub-criteria. These criteria were grouped as the observed persistence of a clone in a population may be the result of a high transmissibility among hosts, of a high persistence in colonized individuals (especially if asymptomatic) and/or from recurrent exposure to a contaminated environment and/or other host species, and thus it may be difficult to disentangle their individual impacts. Our data suggests that the clone A-I might persist on certain farms for 6 months or longer. It is a good illustration of a situation where both environmental persistence and transmission between pigs on the farm may have occurred. Whichever scenario(s) is/are at stake, the clone is still a risk for the swine health and therefore is considered a high risk clone in a One Health context. Nevertheless, these characteristics should be investigated further to determine which mechanism(s) is/are involved to develop fighting strategies. As an example, if persistence in the environment is the main mechanism involved in transmission between batches of pigs, protocols including efficient disinfection should be advised to farms that harbour the clone. On the other hand, if the transmissibility between animals is high, vaccination may be advised. This distinction could not be made with the design of our study. To summarize, the clone A-I has a great capacity of dissemination through either high infectivity or environmental persistence or both and, as a result, has been identified throughout a large territory and therefore fulfils our third criterion to be a high risk clone.

It is noteworthy that isolates from the clones A2, A3 and A4 did not spread as widely as the clone A-I. We believe that the sequence of events, first the acquisition of a higher pathogenicity (isolates from the clone A2 and A3 were identified prior to isolates from A-I) then the acquisition of the resistance to fluoroquinolone, was required to lead to the clone A-I being qualified as high risk. It is not known if the acquisition of fluoroquinolone resistance alone would have allowed the resulting clone to spread as efficiently as the clone A-I through increased fitness mechanisms [10].

To support the critical role of the One Health approach, the inclusion of samples from other origins than pigs, such as environmental samples near the farms and/or human samples such as caretakers, would have been advisable. These samples would have allowed us to assess the potential zoonotic transmission or the persistence in the environment. These two criteria are acknowledged in the multiplicity of sources sub-criterion of our new definition of high risk clone. Indeed, an animal population could be a reservoir for a clone that does not endanger the host population but could threaten public health, such as the Shiga-toxin producing *E. coli* O157:H7. In our case, due to the retrospective nature of our study, environmental and human samples could not be analyzed as they were not available. Therefore, environmental persistence or zoonotic transmission were not evaluated directly. The zoonotic transmission is unlikely, though, as ETEC: F4 are usually not pathogenic for humans.

Most isolates belonging to the clone A-I were MDR, therefore fulfilling the second criterion (Table 1). Indeed, most isolates presented non susceptibility to enrofloxacin, tetracycline and ampicillin. Enrofloxacin non susceptibility was driven by at least 2 mutations in *parC* (S801and E62K) and 2 mutations in *gyrA* (S83L and N652H). In 16 isolates, mostly from the USA, an additional mutation in the *gyrA* gene was detected. The mutation *parC* E62K was detected in all our isolates and has been described in the literature as a “real” mutation [27] although not unanimously [28]. As it was present in all our isolates (including those from the USA), and as isolates only possessing this mutation were susceptible to fluoroquinolones, we considered this mutation as clinically not relevant. The *gyrA* N652H mutation, classified as «unknown» in ResFinder, was present in all isolates from the clonal lineage A (from USA and from Quebec). It is not described in the literature to the authors’ knowledge. Both mutations should be further investigated for MIC levels to evaluate their possible role in sequential acquisition of clinically significant resistance to fluoroquinolones. Ampicillin and tetracycline resistance were driven by *bla_TEM-1_* (142/150) and *tet(A)* (134/150) respectively. In addition, isolates belonging to the clone A-I were associated with several plasmids such as the IncFIB(K) replicon. This plasmid family has been described as “an epidemic plasmid” [29] and is largely associated with specific resistance genes in *Enterobacteriaceae* [30].

*Bla_CMY-2_* is the predominant ESBL/AmpC gene in the ETEC:F4 of swine origin (39/192 isolates) compared to the *bla_CTX-M-1_* (16/192). However, *bla_CTX-M-1_* was mostly identified in isolates from Quebec (15/16). The monitoring of this tendency must be pursued to determine if the *bla_CTX-M_* family is overtaking the rest of the ESBL/AmpC genes in animal populations in North America, as happened in Europe [31]. Nevertheless, the presence of ESBL/AmpC genes in 31 isolates belonging to the clone A-I is concerning as the presence of these genes results in a great capacity to disseminate, thus conferring on the clone another “fitness” advantage. Hence, a subclone carrying systematically such a gene could emerge in the near future, as has occurred for ST131-H30-Rx [32].

Our evaluation of clinical severity was based on retrospective evaluation of non-standardized necropsy reports, which is imperfect. However, it was our only means of evaluating objectively the clinical aspects of increased pathogenicity. We limited our evaluation to one criterion: death of affected pigs on the farm. Although the clone A-I itself was not associated with a higher proportion of death in pigs on the farm compared to other isolates within the clonal lineage A, greater mortality was associated with isolates in the clonal lineage A than that observed with other clonal lineages. Thus, this clone also fulfills the pathogenicity criteria (Table 1). The presence of genes coding for different pili (ECP and HEP) or for T6SS and the presence of known virulence factors such as *ompA* are likely to contribute to an increase in pathogenicity of this clonal lineage. Indeed, each of these elements are known key virulence factors. ECP plays a dual role in early-stage biofilm development and host cell recognition [33]. HCP mediates invasion of epithelial cells or interbacterial connections leading to biofilm formation [34]. T6SS is a multi-protein complex dedicated to the delivery of toxins [35]. Finally, despite its conservation throughout evolution among pathogenic and non-pathogenic bacteria, *ompA* interacts with specific receptors for initiating the pathogenic process in some Gram-negative bacterial infections [36]. However, the presence of these genes alone is not enough to prove that they are responsible for the increased pathogenicity. Further functional studies are needed to ensure the role of each of these elements.

The definition of a clone remains a challenge for all microbiologists. The definition of a bacterial clone, based on whole genome sequencing (WGS) data, should not be based only on a fixed number of SNPs. In the current context, where WGS is being increasingly used as a result of decreased costs, the analytical approaches used in different studies are still subject to a great lack of homogeneity and standardisation because of the many possibilities that they offer. The definition of a clone that we propose, based on the bootstrap values, the number of isolates in the branch and number of SNP differences in relation to the mutation risk adjusted for the period between sampling of isolates, allow reproducibility and minimize over-interpretation of the data. Nevertheless, these criteria are debatable. Firstly, the mutation rate is not constant between bacterial species or even across time within the same species. It is known, particularly in *E. coli*, to depend on the level of stress due to evolution pressure [37]. However, we needed to determine a fixed mutation rate to establish a SNP cut-off which was based on *E. coli* variability during an outbreak [38]. Secondly, the bootstrap value of 1 is very stringent. However, considering the large number of isolates taken into consideration in the study, we chose to be strict to enforce the robustness of our analysis. In another context, for example if fewer isolates are available for the phylogenetic analysis, a bootstrap value of 0.99 or 0.95 might also have been acceptable. Thirdly, we fixed the minimal number of isolates per branch at three to reduce the likelihood of a grouping only due to chance. Finally, although the lag of time between the detection of isolates was taken into account in the definition of a cluster, it could not be considered in the maximum-likelihood approach used to create the phylogeny tree, which represents a limitation.

As observed with all laboratory techniques, there is an inherent error rate associated with whole genome sequencing. However, this is difficult to evaluate considering all the steps of analysis involved in establishing a phylogenetic tree, including the sequencing itself, assembling, multiple alignment, recombination signal removal, or tree building. We evaluated the error rate of our pipeline based on a total of four isolates that were duplicated (for 2 isolates there were 2 duplicates, for 1 isolate there were 3 duplicates, and for 1 isolate there were 4 duplicates). We observed that the error rate was very variable depending on the quality of the raw data. However, for the mean quality of our data, we estimate the error rate to be from 30 to 60 SNP between two duplicates (data not shown). This number strengthens our finding that isolates belonging to the designated clone A-I are very similar, because the SNPs we did detect could be related to this measured error rate.

Although our data demonstrate the capacity of WGS to refine the analysis for detecting clones, they also show that the combination of MLST and serotype data can accurately discriminate clonal lineages. On the other hand, when we compared these results to those obtained with PFGE in our previous study, we observed that only two thirds of the isolates analyzed with both methods were classified correctly by the PFGE when the cut-off of 55% of similarity was used to define a cluster [16]. Price et al. found a similar misclassification for the *E. coli* ST131 [32]. Although PFGE has proven useful in the past, we feel that the use of this method should be limited to specific situations such as outbreak investigations and its results interpreted with great caution, and that PFGE should be replaced by WGS as much as possible.

An additional clone C1, comprising isolates belonging to ST772 and serogroup O23 and having a specific virotype STa:STb:F4 infrequently described in Quebec previously [16], was first observed in 2016. As all these isolates are MDR (non susceptible to tetracycline, TMS and one antimicrobial in the aminoglycoside category), they should be monitored during the coming years to evaluate if they could develop into a high risk clone and represent a threat for porcine health in Quebec and in North America. The absence of quinolone non susceptibility is of note and might represent a fitness disadvantage for this clone.

## 4. Materials and Methods

### 4.1. Proposal of Criteria for a High Risk Clone in a One Health Context and Application

A list of previously used criteria to define a high risk clone for antibiotic resistance dissemination in human medicine was established based on a literature review. This list was used as a starting point for discussion among the study members. These criteria were adapted to fulfill the gaps that the current definition could present for bacterial pathogens as regards to the One Health concept, which could include the addition, removal or updating of criteria. The proposed criteria were then applied to putative ETEC:F4 high-risk clones detected in the study.

### 4.2. Isolate Selection

In a preliminary study, ETEC:F4 isolates, coming from diseased (mostly diarrhea or sudden death) pigs were examined by PFGE as described previously [39]. These isolates were randomly selected from the Animal Pathogenic and Zoonotic *E. coli* (APZEC) database (http://www.apzec.ca/ accessed on 28 February 2021) [16] as follows: 90 isolates from between 1990 and 2012, and 46 or 47 isolates per year from 2013 to 2016 inclusively. A phylogenetic tree based on the PFGE profiles of these 274 isolates was generated using the UPGMA method. The tree was composed of 10 clusters using a 55% similarity cut-off. Between 45 and 55% of isolates were randomly selected from each cluster for whole genome sequencing, for a total of 183 isolates. To follow more recent trends, we also sequenced a random selection of ETEC:F4 isolates in the APZEC database from 2017 (*n* = 24) and 2018 (*n* = 23), that had not been included in the PFGE study.

### 4.3. Antimicrobial Susceptibility

Antimicrobial susceptibility results were extracted from the APZEC database when available. The isolates had been tested for susceptibility to 10 antimicrobial agents using the disk-diffusion (Kirby-Bauer) assay, at either the Animal Health Laboratory of the Ministère de l’Agriculture, des Pêcheries et de l’Alimentation du Québec (MAPAQ) or the Diagnostic Service of the Faculty of Veterinary Medicine, as previously described [16,40]. The antibiotics to be tested were chosen based on clinical relevance. Isolates were considered to be multidrug resistant (MDR) if they were non-susceptible (resistant or intermediate) to at least one antimicrobial in three or more classes of antimicrobial tested [41].

### 4.4. DNA Extraction, Library Preparation and Whole Genome Sequencing

DNA was extracted from an overnight culture of each isolate in LB broth using the QIAamp DNA Mini Kit. Briefly, 1mL of LB broth was centrifuged for each isolate. Then, the pellet was washed with several buffer solutions and ethanol. The supernatant was filtered through the QIAamp Mini spin column (Qiagen, Toronto, ON, Canada) and resuspended with distilled water. All these steps were performed according to the manufacturer’s instructions. The libraries were prepared using the Nextera XT DNA library prep Kit (Illumina, Vancouver, BC, Canada), also according to the manufacturer’s instructions. The isolates were sequenced using Illumina Miseq technology, generating 300 bp paired end reads from libraries. The mean total amount of quality filtered raw sequence was 212.7 MB (minimum 55.23 and maximum 608.49) per isolate and the mean coverage was 61× (minimum 16× and maximum 196× (Appendix A)).

### 4.5. Quality Assessment and Assembly

The Galaxy (https://usegalaxy.org/ accessed on 28 February 2021) platform [42] was used for in-silico analysis. FastQC and MultiQC were used to evaluate the quality of raw data [43,44]. Short-read sequences were assembled using SPAdes (Galaxy Version 3.12.0+galaxy1, Center for Algorithmic Biotechnology, St-Petersburg, Russia) [45], and assembly quality was evaluated using Quast (Galaxy Version 5.0.2+galaxy1, Center for Algorithmic Biotechnology, St-Petersburg, Russia) [46]. An assembly was rejected if the number of contigs was > 400, if the N50 was < 40,000, or if the number of contigs was between 300 and 400 and the N50 < 50,000.

### 4.6. MLST, Serotype, Phylogroup and FimH

MLST [47], O and H serotype [48], and the *fimH* subtype [49] were determined by the analysis of generated FASTA files using the Center of Genomic Epidemiology (CGE) platform (http://www.genomicepidemiology.org/ accessed on 28 February 2021). The *fimH* gene is part of the *fim* operon, which encodes type 1 fimbriae found in most *E. coli* strains. The default parameters were used for each application. Phylogroups were determined with in-silico PCR using the Clermont Typing platform (http://clermontyping.iame-research.center/ accessed on 28 February 2021) [50].

These parameters (MLST, serotype, phylogroup and *fimH* subtype) will be referred to as phylogenetic characteristics.

### 4.7. Virulence and Resistance Gene and Replicon Determination

To determine the presence of virulence and resistance genes and replicons, the blast option in Geneious Prime 2020.1.2 was used with a custom-made comprehensive database. To create this database, FASTA files from ARG-Annot [51], PlasmidFinder [52], VirulenceFinder [53], ResFinder [54,55] and VirulenceFactorDataBase (VFDB) [56] were downloaded. For each individual database, we used the most recent version as of 1 June 2020. The comprehensive database (concatenate, makeblastdb) was then created using the Galaxy platform. Duplicate genes were removed with the tool sRNAtoolbox [57]. The SAS.9.4 software (SAS, SAS Institute Inc., Cary, NC, USA) was used to filter the hits. Hits were eliminated if coverage was less than 60% and homology less than 90%. Then, for each isolate, the best hit for each gene or replicon was selected based on the following criterion, applied sequentially: maximum homology, maximum bit score, minimum e-value, maximum coverage and maximum length. These hits were considered to be present in the isolate.

The presence of chromosomal point mutations was determined using PointFinder on the CGE platform [54].

### 4.8. Phylogenetic Analysis

Initial multiple alignment with the default parameters was performed using CSIphylogeny on the CGE platform [58]. The oldest isolate non-susceptible to enrofloxacin was used as reference. The recombination signal [59] in the SNP output was eliminated using Gubbins [60] on the Galaxy platform (Galaxy Version 0.1.0). The maximum-likelihood phylogenetic tree was generated using FastTree v2.1.10 [61] on the Galaxy platform. The SNP phylogenies were annotated with the relevant metadata using iTOL (http://itol.embl.de accessed on 28 February 2021) [62].

A clonal lineage was defined as a gathering of isolates that belong to the same ST (MLST), the same serogroup, the same phylogroup and same *fimH* gene.

The clones were defined based on three criteria. Firstly, only branches from nodes with a bootstrap value of 1 were retained in the tree. Secondly, a candidate for a clone was only considered if the grouping included three or more isolates. Thirdly, the maximum number of SNPs between pairs of isolates within a group, defined as the SNP_max_ was < M × T × P where M is the mutation rate of *E. coli,* which has been described as 3 × 10^−6^ per year per site [38], T is the number of years between two isolates and P is the number of positions analyzed in all genomes for each phylogenetic analysis. A sub-clone was defined as a clone (defined with the same criteria) within a clone.

Singletons were defined as unique isolates in terms of phylogenetic characteristics (MLST, serotype, *fimH* gene and phylogroup).

### 4.9. Mortality Risk and Stage of Production

The mortality associated with isolates was evaluated for all cases with a necropsy report available from the the Animal Health Laboratory, MAPAQ or the Diagnostic Service of the Faculté de médecine vétérinaire (FMV) of the Université de Montréal. A “case” was defined as a gathering of samples coming from the same farm on the same day, but not necessarily coming from the same animal [16]. Isolates belonging to several pathovirotypes may be found in one case; however, only one isolate belonging to a specific pathotype was randomly selected from a particular case. Each necropsy report was reviewed blinded to the clonal lineage identified. The case was considered as being associated with mortality if the presence of one or more deaths on the farm was reported in the anamnesis or if the pig was submitted dead (either by natural causes or by euthanasia on the farm) for necropsy. Chi square tests were performed to compare the mortality risk between clonal lineages and between clones within the clonal lineage A (see results). When the global comparison with the chi square test was significant (*p* < 0.05), pairwise comparisons between clonal lineages were performed and the Bonferroni correction was applied. As there were few isolates in the clonal lineages D, E and F, these were merged into one category for the analysis.

The phase of production was extracted from the APZEC database for each case and categorized as lactation, weaning, and growing-finishing phase. Exact chi square tests were performed to determine if clonal lineages or clones within the clonal lineage A (see results) were significantly associated with one phase of production. When the global comparisons with the chi square test were significant (*p* < 0.05), pairwise comparisons between clonal lineages were performed and the Bonferroni correction was applied. We also merged isolates in the clonal lineages D, E, and F into one category for this analysis.

The name and location (6-digit postal code) of the farm were also retrieved from the database to assess the persistence of the clone on a farm.

### 4.10. Selection of Other Whole Genome Sequences from GenBank

To evaluate if isolates from other parts of the world cluster with those of the clone detected in Quebec, the Enterobase database was searched (https://enterobase.warwick.ac.uk/ accessed on 28 February 2021) [63]. As the clone A1 (see Figure 1 and results) seemed to have emerged in Quebec in 2013, the period of analysis was restricted to 2013–2018. The search criteria for isolates were as follows: sampled in 2013 or later, in swine, and determined to be from the clonal lineage ETEC ST100, O149H10 and phylogroup A. All isolates corresponding to these criteria were selected and sequences were downloaded from GenBank when available. Phylogenetic analysis was performed as described above, including all isolates (those from Quebec in the present study and those found in GenBank) belonging to the clonal lineage A (see results) sampled from 2013 to 2018.

## 5. Conclusions

In this article, we proposed an adaptation of the high risk clone definition to the One Health concept. Based on this definition, we demonstrated the presence of a high-risk enrofloxacin-non-susceptible ETEC:F4 clone circulating in the pig population in North America. This clone was determined using three well-defined criteria applied to a WGS phylogenetic tree to ensure replicability. The surveillance of this clone and other potential emerging clones is essential. Hence, we need to identify this type of clone to increase our understanding of the conditions that promote their emergence and their dissemination, to implement fighting strategies on the field.

## Figures and Tables

**Figure 1 antibiotics-10-00244-f001:**
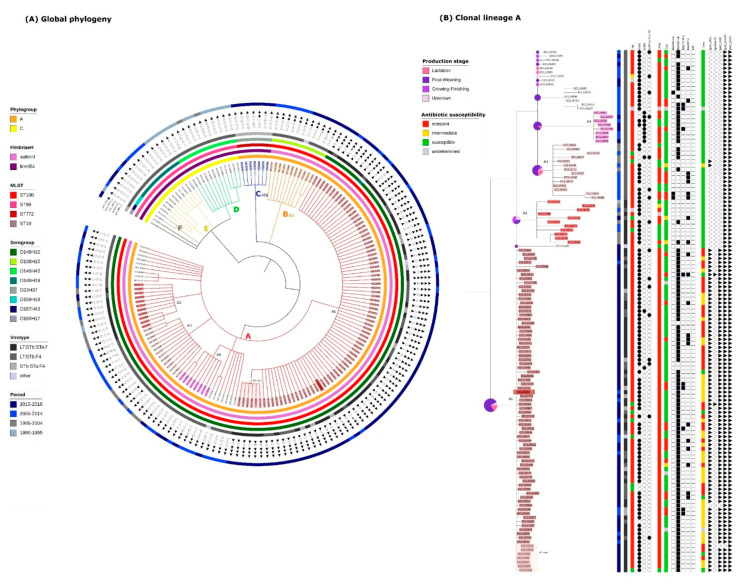
(**A**) Global phylogeny: Circular Tree based on SNP phylogeny. The length of the branches is not proportionate to the phylogenetic distance. Gubbins was used to eliminate the recombination signal. The total SNP number was 19,299. Each branch with a bootstrap value under 1 was collapsed. The capital letters designate each clonal lineage. Filled triangles indicate the presence of the corresponding mutation, empty triangles indicate the absence of the corresponding mutation. (**B**) Clonal lineage A (same phylogeny is used): The number associated with the letter A designates each clone. The identification name for each isolate belonging to a clone is highlighted in a specific color. Circles represent the proportion of the age of the pigs within each corresponding branch. Columns entitled Tet, Amp, Cef, Enro, represent respectively the susceptibility to tetracycline, ampicillin, ceftiofur and enrofloxacin (code for the corresponding color is available on the figure).

**Figure 2 antibiotics-10-00244-f002:**
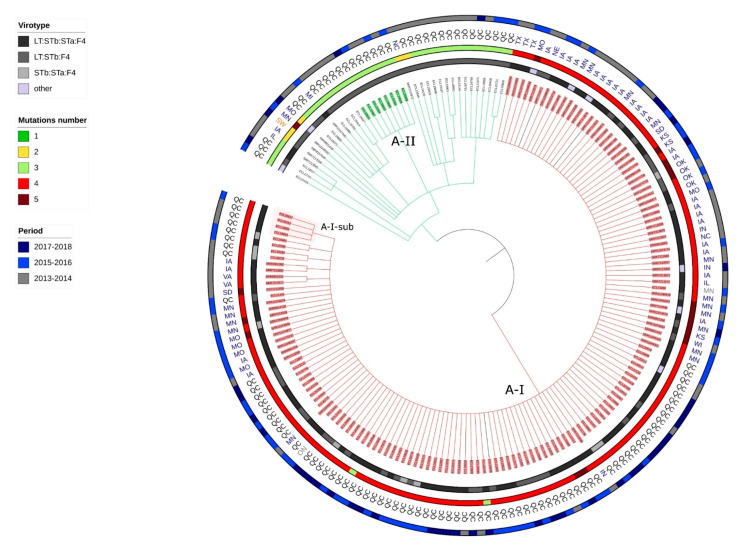
Circular Tree based on SNP phylogeny. The length of the branches is not proportionate to the phylogenetic distance. All isolates belong to the clonal lineage A (ST100-O149H10-phylogroupA-no *fimH* gene) and originated from 2013 or after. Each branch with a bootstrap value under 1 was collapsed. QC = Quebec, SW = Switzerland, MN (in grey) = Manitoba, ON = Ontario, MI = Michigan, MN (in blue) = Minnesota. MO = Missouri, NC = North Carolina, TX = Texas, IA = Iowa, KS = Kansas, WI = Wisconsin, IN = Indiana, IL = Illinois, OK = Oklahoma, NE = Nebraska, SD = South Dakota, VA = Virginia. A-I and A-II designate each clone and A-I-sub designates the subclone within the clone A-I. The 3 isolates colored in black are the three outsiders (more than 70 SNPs of differences in the pairwise distance comparison). The identification name for each isolate belonging to a clone is highlighted in a specific color.

**Table 1 antibiotics-10-00244-t001:** Definition of criteria for determination of a high risk bacterial clone for antibiotic resistance dissemination in the context of the One Health concept.

Criterion	Definition	Application to ETEC:F4 Clone A-I Detected in Pigs in Quebec
**1/Emergence**	The clone is newly recognized, newly evolved or has occurred previously but shows an increase in incidence or expansion in geographical, host or vector range.	The clone A-I emerged in North America in 2013.
**2/Carriage of multiple antimicrobial characteristics**	The clone carries multiple resistance genes associated with phenotypic multidrug resistance. The resistance genes can be carried by mobile genetic elements or by the chromosome (and then results from mutations) or both.	The clone A-I carries at least the genes *tet(A)* and the *bla_TEM-1_* which are associated with phenotypic resistance to tetracyclines and penicillins, respectively. Several replicons such as the IncFII have been identified in the clone A-I. Moreover, it also carries *parC* and *gyrA* mutations responsible for non-susceptibility to fluoroquinolones.
**3/High capacity of dissemination through one or a combination of the following characteristics:**	The clone is likely to disseminate due to:	The clone A-I has been detected in at least one province of Canada as well as in many states of the USA.
**3a/ High infectivity**	- High probability of transmission following exposure to an infected host or environmental source and/or	The clone A-I has been observed in different batches of pigs on the same farm for 6 months.
**3b/ Long-term persistence**	- Long-term persistence and shedding in colonized individuals and/or	
**3c/ Multiplicity of sources**	- Presence of the pathogen in multiple animal species and/or in human population and/or in environmentally maintained source.	
**4/High Pathogenicity**	The clone can cause severe disease in animals or/and in humans.	The clone A-I is associated with a higher risk of mortality that observed for other clonal lineages.

**Table 2 antibiotics-10-00244-t002:** Virulence genes specific of the different clonal lineage (the details are available in Appendix A) * See Appendix A for the complete list of genes.

Genes Clonal Lineage	Enterohemolysin *(ehxA)*	Fimbrial Major Protein *(lfpA)*	Adhesin *(Iha)*	Siderophore Yersiniabactin *	Type VI Secretion System *	*E. coli* Common Pilus (ECP) *	Hemorrhagic *E. coli* Pilus (HEP) *	*ompA* Outer Membrane Protein A	*rpoS* Sigma S (Sigma 38) Factor	*tar/cheM* Methyl-Accepting Chemotaxis Protein II [Peritrichous Flagella]
**A** **(ST100/O149)**	Absent	Absent	Present	Absent	Present	Present	Present	Present	Present	Present
**B** **ST772/O23**	Present	Absent	Absent	Present	Absent	Absent	Absent	Absent	Absent	Absent
**C** **ST100/O138**	Absent	Absent	Present	Present	Absent	Absent	Absent	Absent	Absent	Absent
**D** **ST90/O149H19**	Absent	Present	Absent	Present	Absent	Absent	Absent	Absent	Absent	Absent
**E** **ST90/O157**	Absent	Present	Absent	Present	Absent	Absent	Absent	Absent	Absent	Absent
**F** **ST90/O149H43**	Absent	Present	Absent	Present	Absent	Absent	Absent	Absent	Absent	Absent

**Table 3 antibiotics-10-00244-t003:** Revised criteria for high risk clone applied to the clones identifies in our study.

Criteria/Clone	A-I	A2	A3	A4	B1	C1
**Emergence**	yes	no	no	no	no	yes
**Multidrug resistance**	yes	no	no	no	no	yes
**Potent dissemination**	yes	no	no	no	no	no
**Pathogenicity**	yes	yes	yes	yes	yes	yes

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
