# Peer review of "High Risk Clone: A Proposal of Criteria Adapted to the One Health Context with Application to Enterotoxigenic Escherichia coli in the Pig Population"

_antibiotics, 2021, doi:10.3390/antibiotics10030244_

Round 1
Reviewer 1 Report
Authors sequenced A total of 183 isolates originating from diseased pigs (1990-2018) and detected virulence, resistance genes or replicons. Phylogenetic trees were then constructed and clones were identified using predefined criteria. This is a very interesting work However, some questions need to resolve before publishing:
In Antimicrobial Susceptibility, what are selecting rules when use 10 antimicrobial agents?
Please give the simple description of DNA extraction.
Line 27: have a widespread should be has a widespread…
Line 35: environmental
Line 191, Line 520: please give the full or abbreviated name of MDR.
Line 265: please give the details of farms if it is possible such as location and scale breeding.
Line 290: space format of Figure2 and FigureS2
Line 297: please give italic parC, gyrA and other gene names in full text.
Line 305: the ampC gene
References: non-standard format in Refs. 2, 6, 10, 17-19, 24-26, 32, 47 and 60
Reviewer 2 Report
- The abstract needs to be restructured as background-significance-work done in the paper-salient findings.
- Lines #53-55 need to be rephrased to convey that continued fluoroquinolone treatment despite it resistance amounted to the 'high risk clone'.
- line #62-65 need grammar modification.
- Line #77-please rephrase
- Certain sentences in results need grammar modifications.
- the antibiotic resistance data needs better represented.
- Some of the information seems redundant please take a good look at those and avoid repetitions.
Reviewer 3 Report
The manuscript written by Maud de Lagarde and colleague is a very interesting summary of a study aimed at the revisioning of the criteria for defining as “high risk clone” the bacterial clones that enhance the dissemination of antibiotic resistance. Authors reported/applied the revised criteria suggested to the case of Enterotoxigenic Escherichia Coli (ETEC) F4 Associated with Enrofloxacin Non-Susceptibility in the Porcine Population in North America. The manuscript is to this Reviewer of scientific sound and of a great novelty. Neverteless, to this Reviewer opinion, the One Health aspect is not as central as one might expect from the title and introductive part. In other words, the discussion section of this manuscript should emphasize the importance of revising the high-risk criteria in the one health perspective. To this purpose authors must shed light on the role/importance of the potential transmission of the AMR traits among microorganisms either harboring the same ecological niche (e.g. the gut microbiota) or harboring two apparently distinct niches (e.g. feces and soil). Underlining the potential of transmission of AMR through horizontal gene transfer or other mobile elements (e.g. bacteriophages) across the boundaries of the three One health pillars (human, animals and environment) is of pivotal role in supporting the revised criteria proposed by this manuscript (e.g. doi: 10.3390/ijms21061914). In this light, implementing the experimental design by accounting for samples other than animal derived ones is encouraged, such as considering environmental samples from the neighborhood of the farms and human samples to be taken from the animal caretakers and breeders.
Minor comments:
the title can be shortened to enhance fluency
abstract section should be lightened of the results and better focusing on the main message the study wish to carry and the experimental hypothesis of the study along with the conclusion on the basis of the experiment performed.
Round 2
Reviewer 3 Report
The revised version of the manuscript addressed the issues raised by the Reviewers and it is, in my opinion, suitable for publication.